# Distinctly different bacterial communities in surface and oxygen minimum layers in the Arabian Sea

- 3
- 4 Mandar Bandekar<sup>1</sup>, Nagappa. Ramaiah<sup>1</sup>\*, Anand Jain<sup>1, 2</sup>, Ram Murti Meena<sup>1</sup>
- <sup>5</sup> <sup>1</sup>Microbial Ecology Laboratory, Biological Oceanography Division, CSIR-National Institute of
- 6 Oceanography, Dona Paula, Goa, 403004.
- <sup>7</sup> <sup>2</sup>Present address: Arctic group, National Center for Antarctic and Ocean Research, Sada
- 8 Headland, Vasco-da-Gama, Goa, 403804.
- 9

10 \*Corresponding author: N. Ramaiah (<u>ramaiah@nio.org</u>)

11

# 12 Abstract

13 Contributions of microbial communities to biogeochemical processes in oxygen minimum oceanic zones are being realized through the applications of molecular techniques. To understand 14 seasonal and depth-wise variations in bacterial community structure (BCS) in the Arabian Sea 15 oxygen minimum region, extensive sampling and molecular analyses were carried out. 16S 16 rRNA gene sequencing was done to profile the BCS from five depths, surface (5m), deep 17 18 chorophyll maximum (43-50m, DCM), 250m, 500m and 1000m during Spring intermonsoon (SIM), Fall intermonsoon (FIM), and Northeast monsoon (NEM) seasons. Sequencing of 19 743 chimera-free clones revealed a clear vertical partitioning of BCS between the surface 20 (surface + DCM) and OMZ (250 + 500 + 1000m) layers. There was no distinct seasonal 21 difference in the BCS. Most 16S rRNA gene sequences were affiliated to Gammaproteobacteria 22 (39.31%), Alphaproteobacteria (23.56%) and Cyanobacteria (20.2%). Higher diversity and 23 OTUs in OMZ predominantly consisting of Alteromonodales, Sphinogomonadales, 24 Rhodobacterales, Burkholderales, and Acidimicrobiales we observed might be due to their 25 microaerophilic metabolism, ability to degrade recalcitrant substrates and assimilate sinking 26 particulate matter. Further hitherto undescribed diversity both in surface and OMZ layers was 27 28 evidenced. Implicit role of extant bacterial community in denitrification and anammox and in sulphur oxidation is highlighted. 29

30

Keywords: Oxygen Minimum Zone, Bacterial diversity, 16S rRNA gene, Gammaproteobacteria,

Cyanobacteria, Arabian Sea

# 33 **1 Introduction**

Poor ventilation of intermediate layers, higher microbial respiration and organic matter oxidation within the water column (Wyrtki 1962) lead to oxygen minimum zones (OMZ) with 35 dissolved oxygen (DO) concentrations often below  $< 20\mu M L^{-1}$  (Lisa 2003). Major OMZs are 36 found in the intermediate depths of eastern tropical North Pacific (ETNP, Wyrtki 1966), eastern 37 tropical South Pacific (ETSP, Wyrtki 1966), eastern Arabian Sea (Wyrtki 1973; Madhupartap et 38 al. 1996; Naqvi and Jayakumar 2000) and eastern South Atlantic (Karstensen et al. 2008). Within 39 the OMZ, intense anaerobic and related processing of nitrogenous compounds (Stramma 2008) 40 lead to loss of fixed nitrogen to the atmosphere. Denitrification (Naqvi 1994) and anaerobic 41 oxidation of ammonia (anammox, Dalsgaard et al. 2012) are the major pathways for this nitrogen 42 loss. Hypoxic conditions often select resilient microbes and restrict their vertical distribution 43 (Wishner et al. 1995). 44

The OMZs, earlier believed to occur mostly in nutrient rich upwelling regions, are 46 47 currently expanding and/or intensifying due to anthropogenic impacts (Diaz and Rosenberg 2008). The expansion of OMZs is affecting benthic ecosystems and marine fisheries due to 48 49 habitat alterations and/or changes in nutrient cycling (Stramma et al. 2008). Further, the OMZ expansion is ascribed to increased production of climate active trace gases including carbon 50 dioxide  $(CO_2)$ , methane  $(CH_4)$  and nitrous oxide  $(N_2O)$ . Microbes are involved in all such 51 processes but little is known about their community structure and metabolism within OMZs, in 52 53 particular from the Arabian Sea-OMZ (AS-OMZ).

Coinciding with an active denitrification zone (Naqvi 1994; Naqvi et al. 1998) the AS-55 OMZ is among the largest anoxic regions in the oceans with DO levels  $< 20\mu$ M, sometimes 56 dipping as low as 0.1 µM (Naqvi 2006; Paulmier and Ruiz-Pino 2009). This alone accounts for 57 20% of the oceanic denitrification (Codispoti et al. 2001) and contributes to 40% of the global 58 pelagic dinitrogen (N<sub>2</sub>) production (Naqvi et al. 2008). Previous studies mostly focused on 59 delineating the role of denitrifying and anammox bacteria in the loss of fixed nitrogen in the AS-60 OMZ (Jayakumar et al. 2009; Bulow et al. 2010; Pitcher et al. 2011; Newell et al. 2011; Bouskill 61 et al. 2012). However, detailed attempts to understand bacterial community structure and/or 62 bacterial diversity of the AS-OMZ and its overlying surface waters are a few and far in between 63

(Fuchs et al. 2005; Jain et al. 2014). The present study thus aimed to delineate phylogenetic 65 diversity of the overall bacterial communities in the AS-OMZ, as well as the diversity of bacterial phylotypes contributing to temporally stable bacterial community structure (BCS) in the 66 OMZ (vide Jain et al. 2014). For this, we analyzed small subunit ribosomal RNA gene (16S 67 rRNA or SSU rRNA) clone libraries prepared from water samples collected from five depths 68 from the Arabian Sea Time Series station (ASTS; 17°0.126'N, 67°59.772'E), during three 69 different seasons. ASTS is akin to well known HOTS (Hawaii Ocean Time-Series) in the Pacific 70 Ocean and BATS (Bermuda Atlantic Time-Series) in the Altantic Ocean. 71

# 73 **2 Materials and methods**

# 74 2.1 Sampling

Water samples were collected under the SIBER program in May 2012 (Spring 75 76 intermonsoon [SIM]), September 2012 (Fall intermonosson [FIM]), and February 2013 (Northeast monsoon [NEM]), as described previously by Jain et al (2014). In brief, samples for 77 DNA extractions were collected using pre-cleaned Niskin bottles on a CTD rosette. Samples 78 from surface (0.5m), deep chlorophyll maxima (DCM) (~35-50m), intense denitrification zone 79 80 (250m, 500m) and deep denitrification zone (1000m) at the ASTS station (17°0.126' N, 67°59.772'E) were strained through 200µm pore sized bolting silk, immediately after collection, 81 2.5 L of seawater sample from each depth was filtered peristatically through a Sterivex cartridge 82 fitted with 0.22 µm pore size membrane filter (Millipore, USA). The Sterivex cartridge was then 83 84 filled with 1.8 ml of lysis buffer (50 mM 131 Tris pH 8.3, 40 mM EDTA and 0.75 M sucrose), sealed, and stored frozen at -80°C until nucleic acid extraction was performed in the laboratory. 85 86

#### 00

# 87 2.2 DNA extraction and PCR amplification of 16s rRNA genes

<sup>88</sup> DNA was extracted from Sterivex filters using using the modified method of Ferrari & <sup>89</sup> Hollibaugh (1999). The precipitated DNA was hydrated in 50  $\mu$ l sterile deionised water. All <sup>90</sup> DNA samples were subjected to PCR amplification using universal 16S rRNA primers 27F and <sup>91</sup> 1492R (to confirm that the DNA was of PCR quality. The 16S rRNA gene was amplified <sup>92</sup> following conditions mentioned by Sambrook (1989). PCR amplification was performed in a <sup>93</sup> final volume of 50  $\mu$ l in a thermocycler (Applied Biosystems, USA) and correct amplification <sup>94</sup> was ensured by checking for the amplicons electrophoretically.

#### 95 2.3 Clone library construction and DNA sequencing

PCR amplified 16S rRNA gene products were purified using Axyprep-96 PCR Clean up kit (Axygen, Biosciences), cloned into pCR4-TOPO vector using a TOPO-TA cloning kit for 97 sequencing (Invitrogen, USA) and transformed by chemical transformation into TOP-10 cells as 98 per manufacturer's instructions. As the transformation efficiency was low to moderate, at least 99 thrice cloning trials were repeated to collect a minimum of 65 clones for further analyses. All 100 positive clones/transformants from each sample were picked out, grown overnight at 37°C on LB 101 plates and subjected to the colony-PCR with primers sets pucM13F/pucM13R using temperature 102 conditions as per manufacturer's instructions. PCR products were purified with the Axyprep-96 103 PCR Clean up kit (Axygen, Biosciences) and then sequenced using an ABI 3130XL genetic 104 analyzer (Applied Biosystems, USA). Clone libraries were constructed from particular depths 105 keeping DO profile in mind. However, for the purpose of comparison, clone libraries from 106 surface and DCM depth were considered as surface group (SIM, FIM and NEM). Similarly, 107 108 clone libraries from 250, 500 and 1000m were considered as OMZ group for each season

#### 110 **2.4 Phylogenetic analysis of clone sequences**

The sequences were assembled into contigs using DNA Baser sequence assembly 111 software version 2 (DNA baser, USA). Vector contamination was removed from the sequences 112 using the VecScreen tool (http:// www. ncbi. nlm. nih. gov /tools/vecscreen/). Only consensus 113 sequences without vector and primer residues and with a quality score of 20 (which translates 114 115 into more than 99.5% correct bases, Allex 1999), were used for further analyses. Chimeric 116 sequences were excluded using а chimera detection program (http://decipher.cee.wisc.edu/FindChimeras.html). It is to be noted that close to 15% of the total 117 sequences were chimeric. Phylogenetic affiliation of proper sequences was determined using the 118 119 naïve Bayesian classifier of the RDP sequence classifier tool (Wang et al. 2007) using the 16S rRNA Training Set 26 2015; 120 14 (updated released on May https://rdp.cme.msu.edu/classifier/classifier.jsp). The classification of sequences was done using 121 1,000 pseudo-bootstrap replications at a bootstrap value of 80%, which results in a standard error 122 123 of only 1.3%. The sequences were also compared with other databases including SILVA, NCBI and Greengenes, using MOTHUR. The sequences were assigned to a phylum if their identity was 124 > 90% in the different databases searched. The sequences obtained and described in this study 125

were submitted to the NCBI GenBank database and are available under accession numbers
KJ589647 to KJ590044, KR269603 to KR269693, KR919859 to KR920002, KR919859 to
KR920002 and KR673365 to KR819266.

# 130 2.5 Determination of OTUs

16S rRNA gene sequences were aligned using ClustalX and the alignment was curated 132 using Gblocks (Castresana 2000) to remove poorly aligned positions and divergent regions. 133 Distance matrices were created from the curated alignments with Phylip 3.66. The sequences 134 were then assigned into phylotypes (operational taxonomic units, OTUs) using MOTHUR by 135 applying the average neighbor rule (Schloss and Westcott 2011). 97% cut-off for sequence 136 similarity was used to delimit an OTU.

## 138 **2.6 Estimation of shared and unique OTUs**

In order to elucidate seasonal and depth wise differences in bacterial phylotypes, the fraction of shared and unique OTUs was estimated using MOTHUR. Further, the numbers of shared and unique OTUs between all three seasons surface samples (SIM-surface, FIM-surface, NEM-surface) and all three seasons OMZ samples (SIM-OMZ, FIM-OMZ, NEM-OMZ) were also estimated.

#### 145 **2.7 Construction of Phylogenetic tree**

A phylogenetic tree of representative shared OTUs from the clone libraries was constructed to visualize their relationship and affiliations with the closest relative sequences from the database (Kemble et al. 2011). The tree was constructed as per MEGA-6 using maximum composite likelihood as substitution model and bootstrap values were calculated using neighborjoining method with a resampling size of n = 500.

# 152 **2.8 Statistical analysis of clone libraries**

The clone libraries from the ASTS were analysed statistically using RDP Library Compare (http://rdp.cme.msu.edu/wiki/index.php/Lib\_Compare) and ∫ LIBSHUFF analyses. The Ribosomal Database Project (RDP) II library compare tool uses the naive Bayesian rRNA Classifier Version 1.0 at an 80% confidence threshold (Wang et al. 2007; Cole et al. 2007) to

157 calculate the difference between two libraries. In brief, the RDP LibCompare provides P values 158 for determining statistical significance of abundance differences for individual taxa instead of estimating overall difference between samples. This tool uses the RDP Classifier to assign 159 sequences to taxa. Depending on the abundance of sequences assigned to each taxon, one of two 160 statistical tests is used to compute a P value to determine if a taxon is differentially represented 161 in the libraries. The f LIBSHUFF program was applied to compare bacterial community structure 162 between libraries in a phylogenetic context. It measures differences between communities based 163 on differences between sequences using Monte Carlo test (Schloss et al. 2004; Schloss 2008). ∫ 164 LIBSHUFF implements the Cramer-von Mises statistic to test the generic hypothesis that two 165 communities are the same (Singleton et al. 2001; Schloss et al. 2004). All statistical analyses 166 were performed using MOTHUR software, except RDP. 167

# 169 **2.9 Diversity and richness estimation**

The indices for diversity (Simpson's and Shannon's indices), richness estimates (Jackknife, Chao 1 and ACE), rarefaction and collectors curve were performed using MOTHUR (Schloss et al. 2009). Total richness of the clone libraries was extrapolated from the observed number of OTUs using the three nonparametric richness estimators.

# 175 **3 Results**

# 176 **3.1 Physico-chemical characteristics of the sampling site**

Physico-chemical characteristics (i.e. temperature, salinity, dissolved oxygen, pH, nitrite, nitrate, ammonium, silicate, phosphate and total organic carbon) at the ASTS are described 178 earlier in Jain et al. (2014) and vertical distribution of dissolved oxygen (DO), nitrite (NO<sub>2</sub>) and 179 180 nitrate (NO<sub>3</sub>) during three seasons is shown in Figure 1. In general, surface and DCM depths (43-50 m) are well oxygenated followed by a steep oxycline between DCM and 250 m. The 181 average DO concentration ranged from 185.24  $\pm$  31.1  $\mu$ M L<sup>-1</sup>at DCM to 5.56  $\pm$  5.5  $\mu$ M L<sup>-1</sup> at 182 250m. The DO concentrations were slightly more at 1000 m. The nitrite concentration was 0.29 183 in the upper thermocline region (50m) with higher oxygen concentrations (during FIM) and 2.5 184  $\mu M L^{-1}$  in the intermediate depths (250m) with low oxygen (during NEM). Surface waters 185 186 during SIM and FIM were devoid of nitrate and it was quite low during the NEM.

## 188

# 189 **3.2 Bacterial Community Structure**

Seasonal distribution pattern of clone sequences from surface and OMZ (Figure 2) 190 indicate that maximum numbers of clone sequences were affiliated to three major phylogenetic 191 groups. These are Gammaproteobacteria (39.31%), Alphaproteobacteria (23.56%) and 192 Cyanobacteria (20.2%). These groups represented up to 82% of the usable sequences. The 193 relative proportions of sequences in these groups varied temporally, as well as spatially, with 194 depth. The percentages of Gamma- and Alphaproteobacteria were much higher in the OMZ than 195 in the surface layers. During NEM the percentages of Cyanobacteria and Gammaproteobacteria 196 were the highest at the surface and in the DCM. Notably, the percentage of 197 Gammaproteobacteria within OMZ (250, 500 and 1000m) did not vary much between seasons. 198 The distribution of Alphaproteobacteria in OMZ was quite similar during FIM and NEM. The 199 highest proportion of unclassifiable bacterial sequences in the surface layers, as well as OMZ 200 201 depths, was observed during SIM, followed by FIM and NEM.

# 203 3.3 Shared and Unique OTUs

The shared and unique OTUs between all three seasons in the surface and OMZ samples 204 were divided into four categories: (1) OTUs common to all seasons, (2) OTUs common to two 205 seasons, (3) season-specific OTUs having >1 sequence, and (4) season-specific OTUs having 206 only one sequence (singleton). Three OTUs common to all season's within surface (Common to 207 208 all season's surface-OTUs, CTASS-OTU) was represented by a minor fraction (11-17 %) of the sequences in individual seasons (Figure 3). Further, 20 OTUs (Common to 2 seasons surface, 209 CT2SS-OTU) were shared within surface during two of the three seasons and represented by 19 210 to 38 % of the sequences in individual seasons. The proportion of season-specific OTUs 211 212 (containing only one sequence) in the surface was the largest (43-73%) and represented by 19-57% of sequences in individual season. 213

A total of eight OTUs were common within OMZ (Common to all seasons OMZ, CTASO-OTU) during three seasons and represented by 14-37% of individual season sequences. Further, 20 OTUs (Common to 2 seasons OMZ, CT2SO-OTU) shared within OMZ during two of the three seasons represented 15-20 % of the individual season sequences. The proportion of

season-specific OTUs in the OMZ was the largest (50-63%) and represented by 28-46% of the
individual sequences from any of the seasons. More than half (63%) of season-specific OTUs in
the OMZ are represented by single sequences.

# 223 **3.4 Phylogenetic affiliation and phylogenetic tree of shared OTUs**

The relative abundance and phylogenetic affiliation of the OTUs shared by all seasons 224 and any of the two seasons in surface and OMZ are listed in **Table 1(a)** and **Table 1(b)**. To study 225 the evolutionary differences between shared OTUs, representative sequences were subjected to 226 phylogenetic analysis and the relationship with the database sequences was depicted using a 227 phylogenetic tree for surface (Figure 4) and OMZ (Figure 5). All CTASS-OTUs (CTASS-OTU-228 1, 2 and 3) represented 13% of all surface sequences and were related to the Cyanobacteria, 229 more precisely to the order Synechococcales. On the phylogenetic tree CTASS-OTU-1 and 230 CTASS-OTU-3 appears on the same branch while CTASS-OTU-2 branches independently from 231 232 them and their closest relatives are sequences from Red Sea. The OTUs shared between two of the three seasons (CT2SS-OTUs) were mainly affiliated with the Cyanobacteria, Proteobacteria 233 and Acidobacteria. SIM-Surface and FIM-Surface shared OTUs were related to 234 Gammaproteobacteria (CT2SS-OTU-4, 9, 11 and 13), Betaproteobacteria (CT2SS-OTU-10), 235 Alphaproteobacteria (CT2SS -OTU-14 and 15), Cyanobacteria (CT2SS-OTU-8 and 16) and 236 Acidobacteria (CT2SS-OTU-20). FIM-Surface and NEM-Surface shared ten OTUs, of which six 237 were affiliated to Cyanobacteria (CT2SS-OTU-3, 5, 12, 17 and 18), two to 238 239 Gammaproteobacteria (CT2SS-OTU-2 and 21) and one each to Alphaproteobacteria (SS-OTU-1) and Betaproteobacteria (CT2SS-OTU-19). SIM-Surface and NEM-Surface shared only one 240 OTU affiliated to Gammaproteobacteria (CT2SS-OTU-6). 241

Of the eight CTASO-OTUs, the two most abundant ones, namely CTASO-OTU-1 and CTASO-OTU-2, represented 16% of all OMZ sequences and were related to *Gammaproteobacteria*, in the Order *Alteromonadales*. CTASO-OTU-1 and -2 branched independently from each other and their closest relatives are sequences from Arctic Ocean, Ishigaki Jima Island (Japan), South China Sea and Red Sea. The next most abundant CTASO-OTU-3 and CTASO-OTU-4 represented 5% of all OMZ sequences and belong to the Order *Sphingomonadales* of Class *Alphaproteobacteria*. Both these aligned on the same branch in the

tree and are related to sequences from South Atlantic Ocean and South Pacific gyre. The CTASO-OTU-5, CTASO-OTU-6, CTASO-OTU-7 and CTASO-OTU-8, represented 1% of the total OMZ sequences. They are affiliated with *Burkholderilaes*, *Acidimicrobiales*, uncultured *Gammaproteobacteria*, and *Rhodobacterales*, respectively. The closest relatives of CTASO-OTU-5, 6, 7, and 8 are reported from Sri Lanka thermal spring, Indian Ocean, Red Sea, and Submarine basalt loihi Seamount, respectively.

The OTUs shared in the OMZ between two seasons were mainly from three different 257 Phyla, including Proteobacteria, Actinobacteria and marine group A (MGA). SIM-OMZ and 258 FIM-OMZ were found to share OTUs related to Gammaproteobacteria (CT2SO-OTU-2, 3, 10, 259 11, 12, and 15), MGA (CT2SO-OTU-12 and 15) and Alphaproteobacteria (SO-OTU-11). The 260 FIM-OMZ and NEM-OMZ shared four OTUs. Of those, three were affiliated to 261 Alphaproteobacteria (CT2SO-OTU-1, 14, and 16) and one to Actinobacteria (CT2SO-OTU-20). 262 The SIM-OMZ and NEM-OMZ shared 10 OTUs. Four of them were affiliated with to 263 Alphaproteobacteria (CT2SO-OTU-6, 7, 8, and 18), another four to Gammaproteobacteria 264 (CT2SO-OTU-9, 13, 17, and 20), and one each to MGA (CT2SO-OTU-4) and Actinobacteria 265 (CT2SO-OTU-5). 266

# 268 **3.5 Comparison of clone libraries**

Using the RDP classifier, it was seen that at Class level, the surface samples during all 269 270 three seasons had more representatives of Cyanobacteria than in the OMZ samples. This was entirely different in the OMZ where Alphaproteobacteria, Gammaproteobacteria and 271 Marinimicrobia were the dominant groups (Table 2). Further, from the RDP libcompare it was 272 clear that the abundance of Gammaproteobacteria, (Altermonodales), Alphaproteobacteria 273 274 (Spingomondales), and SAR11 differed significantly between seasons in the surface clone libraries. In contrast, the abundance of Cyanobacteria and Alphaproteobacteria was significantly 275 lower and different in the OMZ (Table 3). The LIBSHUFF (p 

#### 281 **3.6 Analysis of clone diversity and richness**

Shannon and Simpson indices, as well as rarefaction curves, clearly indicate that at an evolutionary distance of 3% the bacterial diversity in both surface and OMZ was the highest 283 during SIM followed by NEM and FIM (Table 5). However, during each season, higher 284 diversity indices were evident in the OMZ than in the surface layers. The nonparametric 285 Jackknife, Chao 1 and ACE estimators also revealed that the estimated OTUs are much higher in 286 the OMZ than the observed number of OTUs during all the seasons. Rarefaction (Figure 6a) and 287 collector curves (Figure 6b) implied that no saturation was reached either at sequence or OTU 288 level at an evolutionary distance of 1% and 3%, respectively. However, at Class or Family level, 289 at distances of 10%, the rarefaction implied some saturation. 290

# 292 **4 Discussion:**

Traditionally the OMZs were seen as regions dominated by heterotrophic denitrification 293 fueled by sinking of organic matter produced via photosynthesis in the sunlit surface ocean. They 294 were also considered to process a fundamentally different microbial community and operate on a 295 different biogeochemistry. The discovery Anammox, and active but cryptic sulfur cycle in 296 anoxic OMZs have significantly shifted the old paradigms. For almost a decade now the OMZ 297 gene surveys have focused extensively on microbes performing denitrification and anaerobic 298 ammonia oxidation (anammox) with less emphasis on the overall microbial community. 299 Microbial communities within OMZs play central roles in ocean, yet we still lack a fundamental 300 301 understanding of how microbial biodiversity is distributed across the OMZs. In this regard, our 302 efforts are useful in providing some novel insights. Arabian Sea is modulated seasonally by upwelling, winter cooling (Prasanna Kumar et al. 2001) and semi-annual reversal of monsoonal 303 winds (Madhupratap et al. 1996). This greatly influences primary production, organic carbon 304 concentration and flux (Hansell and Peltzer 1998) and bacterial abundance (Ramaiah et al. 1996, 305 306 2000; Jain et al. 2014). In spite of the global biogeochemical and climatic importance of AS-OMZ, spatio-temporal variation (Riemann et al. 1999; Fuchs et al. 2005; Jain et al. 2014) and 307 phylogenetic diversity of the microorganisms inhabiting therein are only sparsely addressed 308 (Riemann et al. 1999; Jayakumar et al. 2009; Divya et al. 2010, 2011). Oxygen deficient waters 309 in the intermediate depths is a perennial feature of the Northeastern Arabian Sea which gets 310 intensified during winter or NEM season due to poor water circulation and high surface 311

productivity (Naqvi et al. 1990; Prasanna Kumar and Prasad 1996). Lower DO level and higher nitrite concentration at the intermediate depth during NEM season signifies intense denitrification processes and the presence of nitrite in the thermocline region signifies the nitrification process (Sen Gupta et al. 1976). During intermonsoon (SIM and FIM) periods, the intense solar heating and weak winds stratify the Northeastern Arabian Sea surface layer, leading to depletion of nitrate in the upper euphotic zone (Muraleedharan and Prasanna Kumar 1996).

The SIM is a transition period from winter to summer and is generally known to be a 319 period of low primary productivity caused by water column stratification and oligotrophic 320 conditions (Madhupratap et al. 1996). During this period, the bacterial community is mainly 321 sustained by the slow-to-degrade dissolved organic carbon (DOC) from earlier phytoplankton 322 blooms of the NEM (Ramaiah et al. 2000). Despite these environmental conditions we observed 323 higher bacterial diversity during SIM (both in surface as well as OMZ). Most microbial habitats 324 are spatially heterogeneous (Kassen and Rainey 2004) and can contain a large number of 325 potential niches. Previous studies reported a positive correlation between habitat heterogeneity 326 ("patchiness") and the phylogenetic diversity of bacteria (Korona et al. 1994; Rainey et al. 2000; 327 Kassen et al. 2000; Zhou et al. 2002). We find that the bacterial diversity at the ASTS varies 328 seasonally in the surface layers and in OMZ and is at its highest during SIM. High rates of 329 denitrification might be among the many possible causes for this higher diversity. Further, the 330 high proportion of unclassified bacteria may also contribute to the increased diversity to some 331 332 extent as observed during SIM.

Though over 60 clones were finally available from each depth for sequencing, they do 334 not capture the full bacterial diversity in the AS-OMZ. This notwithstanding, our efforts are 335 336 useful to suggest that overall bacterial diversity at Phylum and OTU level in the ASTS is higher in OMZ than in the surface layers. Interestingly, the depth-dependent variation of OMZ bacterial 337 diversity is not consistent among different studies (Ganesh et al. 2014). Further, Bryant et al 338 (2012) reported a consistent decline in the bacterial diversity within the ETSP-OMZ. Similarly, 339 Zaikova et al (2010) observed low diversity associated with the seasonal OMZ off British 340 Columbia. In contrast, Stevens and Ulloa (2008), based on 16S rRNA clone libraries identified 341 higher OTU diversity at the ETSP-OMZ, a pattern consistent with the observation in our study. 342

Similarly, Brown et al (2009) and Kemble et al (2011) showed elevated OTU richness to 344 coincide with the zone of minimum oxygen at the HOTS. Bryant et al (2012) and Jain et al (2014) attributed lower diversity in the OMZ to competition for limited resources, environmental 345 filtering, and lower redox potential and less readily available organic matter. On the other hand, 346 higher diversity in the OMZ has been linked to the use of a wider range of terminal oxidants 347 348 compared to the oxic depths where oxygen is the dominant electron acceptor (Stevens and Ulloa 2008). Most clades of *Gammaproteobacteria* are known to denitrify using nitrate as electron 349 acceptor (Miller et al. 2010). Among them, the predominant Alteromonodales in the AS-OMZ 350 appears to be a proficient denitrifier group. 351

Bacterial community structure at the ASTS is dominated by phylotypes affiliated with three major phylogenetic groups viz., Gammaproteobacteria, Alphaproteobacteria and 354 Cyanobacteria. The dominance of Alphaproteobacteria and Gammaproteobacteria, has been 355 reported earlier from a variety of pelagic marine environments (Giovannoni and Rappé 2000), 356 including the OMZ of the ETSP (Stevens and Ulloa 2008; Ganesh et al. 2014) and the Southern 357 Arabian Sea (Fuchs et al. 2005). Our results suggest seasonal variation in the abundance of 358 certain major bacterial groups mainly in the surface layers than in the OMZ depths. As Hansell 359 & Peltzer (1998) suggested, the organic carbon concentration and changes in primary 360 productivity patterns in the surface layers may be responsible for variation in the abundance of 361 Gammaproteobacteria (Altermonodales) and Alphaproteobacteria (Spingomondales). Higher 362 363 abundance of Gammaproteobacteria (Altermonodales) and Alphaproteobacteria (Spingomondales) and SAR11 in the OMZ libraries might be supported by microaerophilic 364 metabolism, particle associated lifestyle and/or their ability to use abundantly available nitrate as 365 their terminal electron acceptor for energy generation. 366

367

The total number of common OTUs shared within surface clone libraries was significantly lower than that in the OMZ. All common surface OTUs constitute only a minor fraction (13 %) of the total surface sequences and were mostly affiliated with *Cyanobacteria*, more precisely to the genus *Synechococcus* and *Prochlorococcus*. Interestingly, majority of surface OTUs (shared between two of the three clone libraries) were also affiliated with *Cyanobacteria*, specifically to *Prochlorococcus*. *Cyanobacteria* of the genera *Prochlorococcus*