# Peer review of "Distinctly different bacterial communities in surface and oxygen minimum layers in the Arabian Sea"

_Biogeosciences, 2016_

## Referee Comment (RC1) · Anonymous Referee #1 · 17 Jun 2016

The manuscript by Bandekar et al, entitled 'Distinctly different bacterial communities in surface and oxygen minimum layers in the Arabian Sea' uses 16S rRNA gene sequences to examine seasonal and depth variations in bacterial community structure in the Arabian Sea. The oxygen minimum zone of the Arabian Sea has been relatively poorly studied to date and with the expansion of these regions predicted, an analysis of the bacterial community structure and the drivers of this, makes this manuscript timely. However, I have a number of concerns that I have outlined in detail below, that need to be addressed prior to publication.

The focus of the paper seems a little confusing as it is currently structured. Based on the introduction, beginning of the discussion and last sentence of the abstract, the

[Figure]

authors seem to want to make this a nitrogen story, but have minimal data to support this being the focus of the paper. In the discussion of the data they only discuss denitrification for four lines and anammox is never mentioned. I think the authors should restructure the paper, to focus it on community structure and diversity and the drivers of this. Currently the auxiliary data available is not at all utilized, where it could be used for multivariate statistical analysis or something similar, to potentially reveal what variables are shaping the community, instead of relying purely on the literature.

The authors also need to emphasize how this study builds on the work by Jain et al, 2014 'Temporally invariable bacterial community structure in the Arabian Sea oxygen minimum zone', which was conducted at the same time as this study, by a number of the authors of this manuscript.

The referencing throughout needs urgent attention, a large number of references are missing and / or incorrect ones used.

I think it would be beneficial for the authors to ask a native speaker to read the manuscript for both language and grammar.

Line 16: 'zone' instead of 'region'

Line 28 to 29: This sentence seems unnecessary, your data shows no evidence for anammox (discussed more below) and denitrification is only very briefly touched on with respect to your own dataset.

Introduction: A more detailed introduction to the Arabian Sea is needed here, discussing circulation and the monsoon.

Line 34 to 36: Surely OMZs are a result of both poor ventilation and productivity fuelling the rain down of organic matter, this needs to be more explicitly addressed.

Line 36: Lisa, 2003 is not an appropriate reference here, I suggest Karstensen et al, 2008; Gruber et al, 2007 and / or Codispoti et al, 2001.

[Figure]

Line 40 to 44: I think it is important that you emphasize here the importance of OMZs for global nitrogen loss.

Line 40: What do you mean by 'intense'?

Line 40: Stramma, 2008 is not the appropriate reference for the cycling of N within OMZs.

Line 43: Why are we suddenly discussing hypoxic conditions, this has not been defined and in the previous sentence you are discussing anaerobic pathways.

Line 51: Reference needed.

Line 51 to 53: There are a number of manuscripts that discuss both the community structure and metabolism within OMZs, for example Stewart et al, 2012; Wright et al, 2012 etc. I agree that less is known about the Arabian Sea, but surely the Jain et al, 2014 paper is relevant here.

Line 57 to 59: The way this is currently written is confusing – im assuming you are trying to say something along the lines of 'OMZs account for 20 to 40% of global N loss from the oceans, with 10 to 20% thought to occur in the Arabian Sea' Codispoti et al, 2001; Gruber, 2004 etc

Line 61: The references Newell et al, 2011 and Bouskill et al, 2012 refer to nitrification not N loss processes, so should be deleted. A large number of studies that have looked at N loss in the Arabian Sea are never mentioned Devol et al, 2006; Nicholls et al, 2007; Ward et al, 2009; Jensen et al, 2011; Lam et al, 2011.

Methods: How the oxygen and nutrient data were collected and analyzed is not mentioned, please include.

Line 79 to 80: I find the terms 'intense dentirification zone' and 'deep denitrification zone' misleading and confusing as denitrification is only discussed very briefly with respect to your data in this manuscript. Im assuming this is because you want to

use the same terms as Jain et al, 2014, but upper and lower OMZ would be more appropriate.

Line 105 to 106: What does this sentence mean 'you kept the DO profile in mind'?

Section 3.1: Define your OMZ for the reader. Use values in this section, terms like 'slightly more' and 'quite low' are meaningless to the reader, unless they are relative to something.

Section 3.3: This section is very hard to read. Why in the figure are the OTUs shown as a percentage, but here you mention a number, please be consistent. Please also check all of the values mentioned in this section, as they do not all currently match the figure.

Discussion: subsections here would help the reader. Why does the discussion start with denitrification and anammox when this only takes up four lines of the whole discussion when you start to discuss your own data. Do the authors think the lack of evidence for anammox is a result of saturation not being reached at OTU or sequence level, so rarer organisms such as Planctomycetes are not being picked up? I would restructure here to focus the beginning of the discussion on community structure and diversity. It would be advantageous to compare this to your auxiliary data oxygen, nutrients, TOC, to see what variables are shaping the community.

Line 296 to 297: References needed

Line 297 to 299: This is not true, a number of studies have examined the community in OMZs and you even cite some of them later in the discussion, for example, Ulloa et al, 2006; Bryant et al, 2012; Stewart et al, 2012; Wright et al, 2012 etc

Line 302 to 317: This needs to be moved to the introduction.

Line 329 to 330: What is your evidence for this statement?

Line 338: The variation in diversity with depth in Ganesh et al, 2014 is dependent on

the size fraction looked at, this is an important point that is currently missing from this sentence.

Line 344: The difference to the Jain et al, 2014 manuscript warrants further discussion here, you sampled at exactly the same stations / depths at the same time. In general, why are not more comparisons made to the Jain et al, 2014 manuscript and how your study builds on that work?

Line 351: 'proficient'?

Line 359 to 362: You have TOC data, why are not testing this hypothesis? Why are you not using your auxiliary data to look at what is shaping the community – you never even refer to the oxygen and nutrient data in Figure 1 within the discussion.

Line 362 to 366: References or evidence for this statement is needed.

Line 375: Include references Lavin et al, 2010 and Ulloa et al, 2013.

Line 381: Reference needed.

Line 383: Surely a reference for N cycling in the Arabian Sea would be more appropriate here.

Figure 1: Use the same scale across all panels. Is it really necessary to have the same oxygen profiles in the upper and lower panels?

Figure 4 and 5: Where are the reference sequences?

---

## Author Comment (AC1) · 12 Jul 2016

Dear Dr. Robinson,

Subject: Submission of revised manuscript bg-2016-147

Thank you very much for obtaining excellent set of expert reviews for our submission. As you will kindly see, we have revised the manuscript in accordance with the advices offered. We have addressed all the points suggested and list our replies in 'Responses to reviewers' comments'.

I append below the replies to all comments we received for the ms.

[Figure]

I hope that these changes are acceptable to you. We believe that with these changes, the revised version meets the requirements and you will kindly accord acceptance to our submission.

I look forward to your kind and early response.

With my Best Regards.

Yours sincerely,

N. Ramaiah Biological Oceanography Division CSIR-National Institute of Oceanography Dona-Paula, Goa, India

Responses to reviewers' comments

Comment: The manuscript by Bandekar et al, entitled 'Distinctly different bacterial communities in surface and oxygen minimum layers in the Arabian Sea' uses 16S rRNA gene sequences to examine seasonal and depth variations in bacterial community structure in the Arabian Sea. The oxygen minimum zone of the Arabian Sea has been relatively poorly studied to date and with the expansion of these regions predicted, an analysis of the bacterial community structure and the drivers of this, makes this manuscript timely. However, I have a number of concerns that I have outlined in detail below, that need to be addressed prior to publication. Reply: We are most grateful for the excellent set of review suggestions you have offered to improve our manuscript. We have strived and accordingly revised the manuscript and complied with all the advices provided.

Comment: The focus of the paper seems a little confusing as it is currently structured. Based on the introduction, beginning of the discussion and last sentence of the abstract, the authors seem to want to make this a nitrogen story, but have minimal data to support this being the focus of the paper. In the discussion of the data they only discuss denitrification for four lines and anammox is never mentioned. I think the authors should restructure the paper, to focus it on community structure and diversity and the

drivers of this. Reply: This advice is carefully addressed by revising the introduction almost entirely and suitably altering the discussion. Discussion involving very sparse communities of denitrification and anammox are pointed out. Indeed, this surprising result was highlighted earlier itself. More details provided in revision in lines 447-453 of the revised version as follows:. "It is interesting to note that only a few of the common OMZ OTUs affiliated to known denitrifiers, suggesting that denitrification is spatially heterogenous (niche segregation). We found just two 16S rRNA clones of Plancto-mycetes (NCBI Accession numbers KJ589907, KJ589971) during just one seasons. We therefore suppose that anammox does occur. Lack of larger numbers of clones Planctomycetes which is a known anammox bacterium may be a result of saturation not being reached at OTU or sequence level. Thus some rarer organisms were not being picked up even after repeated cloning efforts."

Comment: Currently the auxiliary data available is not at all utilized, where it could be used for multivariate statistical analysis or something similar, to potentially reveal what variables are shaping the community, instead of relying purely on the literature. The authors also need to emphasize how this study builds on the work by Jain et al, 2014 'Temporally invariable bacterial community structure in the Arabian Sea oxygen minimum zone', which was conducted at the same time as this study, by a number of the authors of this manuscript. Reply: Thanks a lot for this valuable suggestion. As you will kindly see, in the revision, we have added multivariate analysis in methods (Please see lines 201-205 in the revised version ) Results (Please see lines 320-328 Figure 6 and Table 5 in the revised version) and discussion (Please see lines 379-383 in the revised version) sections. All these sections are reproduced later on in this reply itself.

Comment: The referencing throughout needs urgent attention, a large number of references are missing and / or incorrect ones used. Reply: All suggested references replaced.

Comment: I think it would be beneficial for the authors to ask a native speaker to read the manuscript for both language and grammar. Reply: We are extremely sorry for

the long sentences with conjunctions (many "ands") in the previous version. Such long statements surely made the reading often awkward and laborious. I am sorry for this. This concern is now taken care of, hopefully very largely.

Comment: Line 16: 'zone' instead of 'region' Reply: The word 'region' is replaced by 'zone' (It will in line 16 in the revised version)

Comment: Line 28 to 29: This sentence seems unnecessary, your data shows no evidence for anammox (discussed more below) and denitrification is only very briefly touched on with respect to your own dataset. Reply: Lines 27, 28 rephrased as follows: Hitherto undescribed diversity of denitrifiers, anammox and sulfur metabolizing bacteria was evidenced.

Comment: Line 34 to 36: Surely OMZs are a result of both poor ventilation and productivity fuelling the rain down of organic matter, this needs to be more explicitly addressed Reply: In fact, most of the introduction section is rewritten keeping your advice in the fore. The Introduction now reads as follows: "Intermediate depths in the oceans with very low oxygen saturations of $< 20\mu$M L-1 (Karstensen et al. 2008,) are known as oxygen minimum zones (OMZs). These suboxic layers are usually associated with low oxygen supply but high oxygen demand. This is due to poor ventilation of intermediate depths, higher microbial respiration and organic matter oxidation (Wyrtki 1962, Karstensen et al. 2008,). Major OMZs are found in the intermediate depths of eastern tropical North Pacific (ETNP, Wyrtki 1966), eastern tropical South Pacific (ETSP, Wyrtki 1966), eastern Arabian Sea (Wyrtki 1973, Madhupartap et al. 1996, Naqvi and Jayakumar 2000) and eastern South Atlantic (Karstensen et al. 2008). The OMZs, believed earlier to occur mostly in nutrient rich upwelling regions, are currently expanding and/or intensifying due to anthropogenic impacts (Diaz and Rosenberg 2008). Their expansion is affecting pelagic fisheries and benthic ecosystems due to habitat alterations and/or changes in nutrient cycling (Stramma et al. 2008). Further, the OMZ expansion is ascribed to increased production of climate active trace gases including carbon dioxide ($CO_2$), methane ($CH_4$) and nitrous oxide ($N_2O$, Loscher et al. 2015).

Pelagic environments with low oxygen are responsible for ∼35% of the marine-fixed nitrogen loss through microbially catalyzed reductive processes (Codispoti 2007). These are the regions of nitrous oxide production, a potent greenhouse gas. The Arabian Sea (AS), a biologically most productive tropical basin, is among the largest suboxic regions in the world oceans. A perennial feature in its northeastern region is the oxygen deficient waters in the intermediate depths (∼150-1000m column). Characteristically, the seasonally reversing southwest (SWM: June- Septemeber) and northeast monsoons (NEM: December-February), offshore upwelling and winter cooling fuel high biological productivity (Madhupratap et al. 1996) and organic carbon production (Hansell and Peltzer 1998). Studies by Naqvi et al. (1994), Naqvi (1999) have led to recognize extensive denitrification within the OMZ layers as the main cause for the loss of photosynthetically fixed nitrogen, in particular during SWM and NEM. Accounting for 20% of the global oceanic denitrification (Codispoti et al. 2001), the AS-OMZ alone is responsible for removal of up to 60 Tg of nitrogen annually (Codispoti 2007). Naqvi et al. (2008) estimate that the AS denitrification contributes to ∼40% of the global pelagic N loss. Within the OMZ, microorganisms capable of denitrification and anaerobic ammonium oxidation (anammox) both convert oxidized forms of nitrogen to gaseous species and thereby cause net loss of fixed N. This loss is indicated by the presence of a broad secondary nitrite maximum (SNM) and large fixed N deficits (Naqvi 1991). Apparently, the role of microbial communities in the AS-OMZ N-loss is vital for elucidating the microbes-mediated denitrification in the context of global biogeochemical and climatic processes (Stewart et al. 2012, Ulloa 2012 but little is known about their community structure and metabolism from the AS-OMZ (Jain et al. 2014). Studies by Ramaiah et al. (1996, 2000), Riemann et al. (1999); Fuchs et al. (2005), Jayakumar et al. (2009); Divya et al. (2010, 2011) and Jain et al. (2014) are useful to recognize the heterotrophic bacterial production and abundance and, some of their phylotypes. We lack however, a fundamental understanding of how microbial biodiversity is distributed across the AS- OMZ although microbial communities within OMZs are involved in the biogeochemical processes therein. However, information on spatio-temporal variation

and phylogenetic diversity of bacteria from the AS-OMZ is quite sparse. Coinciding with an active denitrification zone (Naqvi 1994, Naqvi et al. 1998) the AS-OMZ experiences dissolved oxygen (DO) levels sometimes as low as 0.1 $\mu$M (Naqvi 2006, Paulmier and Ruiz-Pino 2009). Previous studies mostly focused on delineating the role of denitrifying and anammox bacteria in the loss of fixed nitrogen in the AS-OMZ (Jayakumar et al. 2009, Bulow et al. 2010, Pitcher et al. 2011, Devol et al. 2006, Ward et al. 2009, Lam et al. 2011). However, detailed analyses to understand bacterial community structure and/or bacterial diversity of the AS-OMZ and its overlying surface waters are a few and far in between (Fuchs et al. 2005, Jain et al. 2014). In the present study we therefore aimed to assess the phylogenetic diversity of the extant bacterial communities in the AS-OMZ. For this, we analyzed small subunit ribosomal RNA gene (16S rRNA or SSU rRNA) clone libraries prepared from water samples collected from five depths from the Arabian Sea Time Series station (ASTS; 17°0.126'N, 67°59.772'E), during three different seasons. ASTS is akin to well-known HOTS (Hawaii Ocean Time-Series) in the Pacific Ocean and BATS (Bermuda Atlantic Time-Series) in the Atlantic Ocean." Comment: Line 36: Lisa, 2003 is not an appropriate reference here, I suggest Karstensen et al, 2008; Gruber et al, 2007 and / or Codispoti et al, 2001 Reply: As per the suggestion, the reference Lisa 2003 is replaced by Karstensen et al. 2008 (It will be in line 35 in the revised version)

Comment: Line 40 to 44: I think it is important that you emphasize here the importance of OMZs for global nitrogen loss. Reply: The importance of OMZs for global nitrogen loss is reviewed comprehensively (lines 59-66 in the revised version) as follows: "Accounting for 20% of the global oceanic denitrification (Codispoti et al. 2001), the AS-OMZ alone is responsible for removal of up to 60 Tg of nitrogen annually (Codispoti 2007). Naqvi et al. (2008) estimate that the AS denitrification contributes to ∼40% of the global pelagic N loss. Within the OMZ, microorganisms capable of denitrification and anaerobic ammonium oxidation (anammox) both convert oxidized forms of nitrogen to gaseous species and thereby cause net loss of fixed N. This loss is indicated by the presence of a broad secondary nitrite maximum (SNM) and large fixed N deficits

(Naqvi 1991)." Comment: Line 40: What do you mean by 'intense'? Reply: Sentence suitably revised (lines 63-65 in the revised version) as follows. "Within the OMZ, microorganisms capable of denitrification and anaerobic ammonium oxidation (anammox) both convert oxidized forms of nitrogen to gaseous species and thereby cause net loss of fixed N." Comment: Line 40: Stramma, 2008 is not the appropriate reference for the cycling of N within OMZs. Reply: Reference of Stramma, 2008 deleted

Comment: Line 43: Why are we suddenly discussing hypoxic conditions, this has not been defined and in the previous sentence you are discussing anaerobic pathways. Reply: Line 43 is deleted.

Comment: Line 51: Reference needed. Reply: Reference of Loscher et al. 2015 added. It will be in line 47 in the revised version

Comment: Line 51 to 53: There are a number of manuscripts that discuss both the community structure and metabolism within OMZs, for example Stewart et al, 2012; Wright et al, 2012 etc. I agree that less is known about the Arabian Sea, but surely the Jain et al, 2014 paper is relevant here. Reply: Reference added. Please see lines 66-70 of revised version which read: "Apparently, the role of microbial communities in the AS-OMZ N-loss is vital for elucidating the microbes-mediated denitrification in the context of global biogeochemical and climatic processes (Stewart et al. 2012, Ulloa, 2012 but little is known about their community structure and metabolism from the AS-OMZ (Jain et al. 2014)." Comment: Line 57 to 59: The way this is currently written is confusing – im assuming you are trying to say something along the lines of 'OMZs account for 20 to 40% of global N loss from the oceans, with 10 to 20% thought to occur in the Arabian Sea' Codispoti et al, 2001; Gruber, 2004 etc. Reply: The sentence is revised. Please see lines 59-62 in the revision which read: "Accounting for 20% of the global oceanic denitrification (Codispoti et al. 2001), the AS-OMZ alone is responsible for removal of up to 60 Tg of nitrogen annually (Codispoti 2007). Naqvi et al. (2008) estimate that the AS denitrification contributes to ∼40% of the global pelagic N loss." Comment: Line 61: The references Newell et al, 2011 and Bouskill et al, 2012 refer

to nitrification not N loss processes, so should be deleted. A large number of studies that have looked at N loss in the Arabian Sea are never mentioned Devol et al, 2006; Nicholls et al, 2007; Ward et al, 2009; Jensen et al, 2011; Lam et al, 2011 Reply: References Newell et al, 2011 and Bouskill et al, 2012 are deleted and replaced with suggested ones of Devol et al. 2006, Ward et al. 2009, Lam et al. 2011 (in line 81 in the revised version). Comment: Methods: How the oxygen and nutrient data were collected and analyzed is not mentioned, please include. Reply: Measurement of Nutrients and dissolved oxygen has been explained in subsection 2.2 of methods section (Lines 107-111 in the revised version) as follows: "2.2 Measurement of nutrients and dissolved oxygen Collection and measurements of physico-chemical parameters from each depth/sample were described in detail earlier in Jain et al. (2014). In brief, the Winkler titration method was followed to measure DO (Carpenter 1965) and standard methods of Grasshoff et al. (1983) to measure the concentrations of ammonia, nitrate, nitrite, phosphate, silicate."

Comment: Line 79 to 80: I find the terms 'intense dentirification zone' and 'deep denitrification zone' misleading and confusing as denitrification is only discussed very briefly with respect to your data in this manuscript. Im assuming this is because you want to use the same terms as Jain et al, 2014, but upper and lower OMZ would be more appropriate. Reply: The terms 'intense dentirification zone' and 'deep denitrification zone' substituted with 'upper OMZ' and 'lower OMZ' (It will be in lines 99 and 100 in the revised version).

Comment: Line 105 to 106: What does this sentence mean 'you kept the DO profile in mind'? Reply: The sentence is rephrased (Please see line 134 in the revised version) as: "Clone libraries were constructed for each sample." Comment: Section 3.1: Define your OMZ for the reader. Use values in this section, terms like 'slightly more' and 'quite low' are meaningless to the reader, unless they are relative to something. Reply: Measured values now included in the text (lines 215-218 in the revised version) as: "The average DO concentrations at 1000m increased to $7.22 \pm 4.5$ $\mu$M L$-1$. The

nitrite concentration was 0.29 $\mu$M L$-1$ in the upper 50m and 2.5 $\mu$M L$-1$ at 250m. Surface waters were almost devoid of nitrate excepting during NEM though with a concentration as low as 0.006 $\mu$M L$-1$." Comment: Section 3.3: This section is very hard to read. Why in the figure are the OTUs shown as a percentage, but here you mention a number, please be consistent. Please also check all of the values mentioned in this section, as they do not all currently match the figure. Reply: Subsection 3.3 of results section is restructured and improved upon. The values in the text now match the Figure 3 now redrawn for an easy understanding. (lines 234-248 in the revised version): "3.3 Operational taxonomic units From the total of 335 clones sequenced from surface samples, as many as 177 OTUs were formed. Among these, only three OTUs were common for all three seasons (Table 1(a)). They covered $\sim$11-17 % of the sequences from individual seasons (Figure 3). Further, of the 177 OTUs, 20 common during any two seasons are also listed in Table 1(a). These covered 19-38 % of the sequences in individual seasons. The proportion of season-specific OTUs generated by just a single sequence was the largest in the surface (43-73%) covering anywhere from 19 to 57% of sequences in any given season (Figure 3). Within the OMZ, eight OTUs were common for all three seasons (Table 1(b)). They covered 14, 24, and 34% of sequences during NEM, FIM, and SIM respectively (Figure 3). Twenty OTUs that covered 15-20% of the sequences were common to any two seasons (Table 1(b)). The proportion of season-specific OTUs in the OMZ was the largest (50-63%) and represented by 28-46% of individual sequences. More than half, or 63%, of season-specific OTUs in the OMZ were single tones (Figure 3; Pl see attached pdf file named: Bandekar et al Fig3)." Comment: Discussion: subsections here would help the reader. Why does the discussion start with denitrification and anammox when this only takes up four lines of the whole discussion when you start to discuss your own data. Do the authors think the lack of evidence for anammox is a result of saturation not being reached at OTU or sequence level, so rarer organisms such as Planctomycetes are not being picked up? I would restructure here to focus the beginning of the discussion on community structure and diversity. It would be advantageous to compare this to your auxiliary data oxygen,

nutrients, TOC, to see what variables are shaping the community. Reply: Subsections are now made in the discussion. The first paragraph is an implicit prelude to the introduction. In fact, we are very grateful to the question on "Do the authors think the lack of evidence for anammox is a result of saturation not being reached at OTU or sequence level, so rarer organisms such as Planctomycetes are not being picked up?" We found only two 16S rRNA clones of Planctomycetes during just one seasons. We therefore suppose that anammox does occur. This statement now included along with the NCBI accession numbers for our Planctomycetes clones. Lines 447-453 in the revised version read: "It is interesting to note that only a few of the common OMZ OTUs affiliated to known denitrifiers, suggesting that denitrification is spatially heterogenous (niche segregation). We found just two 16S rRNA clones of Planctomycetes (NCBI Accession numbers KJ589907, KJ589971) during just one seasons. We therefore suppose that anammox does occur. Lack of larger numbers of clones Planctomycetes which is a known anammox bacterium may be a result of saturation not being reached at OTU or sequence level. Thus some rarer organisms were not being picked up even after repeated cloning efforts."

This said, unlike earlier studies targeting only denitrifiers and/anammox, this study is useful to recognize the extant diversity of bacterial communities in the surface layers and OMZ depths of the AS.

Comment: Line 296 to 297: References needed Reply: Reference of Stewart, 2011 added; line 336 in the revised version

Comment: Line 297 to 299: This is not true, a number of studies have examined the community in OMZs and you even cite some of them later in the discussion, for example, Ulloa et al, 2006; Bryant et al, 2012; Stewart et al, 2012; Wright et al, 2012 etc Reply: Reply: Thanks a lot again. This section is revised thoroughly (lines 336-338 in the revised version) as. "For almost a decade now, the AS-OMZ gene surveys have focused extensively on microbes performing denitrification and anammox with less emphasis on the overall microbial community."

Comment: Line 302 to 317: This needs to be moved to the introduction. Reply: Introduction is thoroughly revised in view of the suggestions, the suggested paragraph from discussion is moved into introduction but has undergone revision as was necessary (Please see above).

Comment: Line 329 to 330: What is your evidence for this statement? Reply: The statement is an inference proposed by us based on prior studies carried out around the ASTS. Lines 348-351 in the revised version read: "We therefore propose that the high rates of denitrification might be among the many possible causes for this higher diversity in the OMZ as also observed by Ramaiah et al. (2009). They reported both higher heterotrophic bacterial production and abundance from the locations surrounding ASTS." Comment: Line 338: The variation in diversity with depth in Ganesh et al, 2014 is dependent on the size fraction looked at, this is an important point that is currently missing from this sentence. Reply: The reference and related text deleted.

Comment: Line 344: The difference to the Jain et al, 2014 manuscript warrants further discussion here, you sampled at exactly the same stations / depths at the same time. In general, why are not more comparisons made to the Jain et al, 2014 manuscript and how your study builds on that work? Reply: A comparison of our work to that of Jain et al, 2014 is now explained in lines 367-370 as "It is to be noted however that previously, as reported in Jain et al. (2014), we examined the composition of bacterial communities through DGGE and by sequencing only 20 prominent and disparate DGGE bands. This was unlike building numerous clone libraries as done for this study."

Comment: Line 351: 'proficient'? Reply: Word has been replaced by 'major' (line 372 in the revised version).

Comment: Line 359 to 362: You have TOC data, why are not testing this hypothesis? Why are you not using your auxiliary data to look at what is shaping the community – you never even refer to the oxygen and nutrient data in Figure 1 within the discussion. Reply: Multivariate analyses was carried out to check the role of auxiliary parameters and described in methods (in subsection 2.10, lines 20-205 in the revised version) and results (These will be in subsection 3.7; lines 320-328 Figure 6 and Table 5 in the revised version). Brief discussion on the same is added (in lines 379-383 in the revised version) as follows: "2.10 Multivariate analyses Principal component analysis (PCA) was performed using Primer 6 (PRIMER-E, Plymouth, UK) to infer relationships between hydrographic parameters and sampling depths during different seasons. All data were normalized to reduce the importance of minor differences in relative concentration/s of individual parameters. 3.7 Principal component analysis (PCA) Based on the Scree plot, the first three component was retained which explained 94.6% of the variation in the environmental data (Table 5; PI see attached pdf named: Bandekar et al Table5nFig6). The sum of all eigenvalues indicated an overall variance of 6.61%. Close to 82% variance in the dataset was accounted for by the first two principal components. The ordination plots distinctly separated the surface and DCM (<50 m) concentrations of DO and TOC from those in the OMZ (250m, 500m, and 1000m; Figure 6). In the overall, clustering together of silicate, nitrate and phosphate concentrations from OMZ depths is discernible. Ammonia and nitrite concentrations from SIM-500 and NEM-250 did not cluster with any of the other seven parameters." In Discussion Section: The available data from the AS-OMZ depths suggest seasonally invariant concentrations of DO, TOC, NO3, NH4 as well as NO2 (Jain et al. 2014). The multivariate analysis brings forth that various hydrographic parameters cluster distinctly as surface or OMZ bunch. It is thus likely that the bacterial community adapted to this steady-state niche ought to remain similar, perennially." Comment: Line 362 to 366: References or evidence for this statement is needed. Reply: Reference of Martinez et al. 2008 added (line 391 in the revised version).

Comment: Line 375: Include references Lavin et al, 2010 and Ulloa et al, 2013 Reply: These references added in line 401 in the revised version).

Comment: Line 381: Reference needed. Reply: Reference of Moore et al. 1998, Moore and Chisholm 1999, West and Scanlan 1999, Rocap et al. 2002, Scanlan et al.

2009 added (line 407 in the revised version).

Comment: Line 383: Surely a reference for N cycling in the Arabian Sea would be more appropriate here. Reply: Reference of Bange et al. 2005 added in line 410 in the revised version.

Comment: Figure 1: Use the same scale across all panels. Is it really necessary to have the same oxygen profiles in the upper and lower panels? Reply: As suggested Figure 1 is revised and same scales are used across all panels (pl see file named: Bandekar et al Fig1.pdf). Since we are comparing nitrate and nitrite versus DO, we believe that inclusion of DO in both panels helpful for a quick discernment. An explanatory sentence for this is included in the revision as "DO concentrations are shown in both panels for easy comparison between DO versus nitrate and nitrite concentrations" in lines 1038-39.

Comment: Figure 4 and 5: Where are the reference sequences? Reply: The reference sequences are all the non-bold NCBI accession numbers used in the trees. A statement now added in lines 1079-1080 and 1087-1088 as: Representative sequences from this study used for constructing the tree are in bold type. All the rest are reference sequences from the NCBI data base.

[Figure]

[Figure]

**Figure 1** Vertical distribution of dissolved oxygen (DO), nitrate ($NO_3$) and nitrite ($NO_2$) concentrations during spring intermonsoon (SIM), fall intermonsoon (FIM), and northeast monsoon (NEM) at the Arabian Sea Time Series (ASTS) location. DO concentrations are shown in both panels for easy comparison between DO *versus* nitrate and nitrite concentrations.

**Fig. 1.**

[Figure]

**Figure 3** Percentage of OTUs and sequences in the surface and OMZ clone libraries obtained during spring intermonsoon (SIM), fall intermonsoon (FIM), and northeast monsoon (NEM) at the Arabian Sea Time Series (ASTS) location."

**Fig. 2.**

**Table 5**: Eigen values, variance percentage of correlation coefficients between environmental variables from surface layers and OMZ depths during Spring intermonsoon (SIM), Fall intermonsoon (FIM), and Northeast monsoon (NEM) at the Arabian Sea Time Series (ASTS) location.

| Variable | PC1 | PC2 | PC3 |
|---|---|---|---|
| Eigenvalues | 4.51 | 1.21 | 0.898 |
| % Variation | 64.5 | 17.3 | 12.8 |
| DO | **0.441** | -0.053 | 0.218 |
| Nitrate | **-0.45** | 0.107 | 0.088 |
| Nitrite | -0.06 | -0.674 | **-0.685** |
| Phosphate | **-0.467** | 0.003 | 0.013 |
| Silicate | **-0.413** | -0.043 | 0.299 |
| Ammonia | 0.052 | **0.726** | -0.606 |
| TOC | **0.456** | -0.049 | 0.139 |

Figures in bold indicate significant values at $p < 0.001$

**Figure 6:**

[Figure]

**Figure 6.** Principal component analysis of environmental parameters from the ASTS. Right hand side circle shows the clustering of hydrographic parameters from the surface samples and the one on left hand side, the clustering from the OMZ depths. Unclustering parameters from SIM-500 and NEM-250 are outside.
**SIM** - spring intermonsoon, **FIM** - fall intermonsoon, **NEM** - northeast monsoon.
**S** - surface, **DCM** - deep chlorophyll maxima
**PC1** - principal component1,  **PC2** - principal component 2

**Fig. 3.**

---

## Referee Comment (RC2) · Anonymous Referee #2 · 3 Aug 2016

The authors have used DNA sequence analysis of environmental 16S rRNA genes and phylogenetic reconstruction to investigate the community diversity of bacteria present at a time-series station in the Arabian Sea. Samples were obtained from the surface mixed layer and from within the oxygen minimum zone (OMZ) on three separate occasions over the course of several months and under contrasting hydrographic conditions. The authors conclude that while there was no distinct seasonal difference in community structure (lines 21-22), greater diversity and community richness were evident within the OMZ when compared to the surface and deep chlorophyll maximum.

While the latter observation is of interest, it is based on a very modest number of sequences from each depth and time point (Table 5) and not that well supported by classical indices of diversity (Shannon and Simpson indexes). As the authors acknowledge at several points in the manuscript (e.g., Section 3.6, lines 334-335, lines 436-438, etc.) their findings are preliminary because due to the under-sampling of the community they have not captured or analyzed in sufficient depth the far greater diversity evidently present at the station in order to draw robust conclusions.

Where they may be on firmer ground is in reporting that the composition of the surface community was distinct from that of the OMZ (Section 3.5). Much of this difference appears to be explained by the absence of cyanobacteria from the deeper samples (lines 195-198, lines 269-270), however, which is an expected result given their photoautotrophic nature. Indeed, if one ignores the contribution of cyanobacteria to the surface community, the relative percentage contributions of the dominant heterotrophs (alpha and gamma proteobacteria) at all depths would be much less distinct than that shown in Figure 2. Reanalyzing the data (minus cyanobacteria) might proof useful.

Minor points:

Line 13 What does 'Contributions" mean? Numbers, biomass, activity?

Line 84 What is 131 Tris?

Section 2.2 Much more detail of the PCR conditions is required. The reference to the manual of Sambrook (et al.) is insufficient - what were the temperatures, times, cycle numbers, etc.?

Section 2.3 TA cloning is very efficient. What explains the poor numbers of transformants recovered in this study?

Line 103 Manufacturers name and reaction conditions required

Line 104-05 Which primers and reaction conditions were used for sequencing?

---

## Author Comment (AC2) · 23 Aug 2016

Dear Dr. Robinson,

Subject: Submission of revised manuscript bg-2016-147

Thank you very much for obtaining excellent set of expert reviews also by Reviewer 2 for our submission. We gratefully thank Reviewer#2 for all the improvement suggestions. As you will kindly see, we have revised the manuscript in accordance with the advices offered. We have addressed all the points suggested and list our replies in 'Responses to reviewers' comments'.

I append below the replies to comments of Reviewer 2 we received for the ms. To

comply with, the suggestions by Reviewer 2, we have now added some text in response to the revision/inclusions. This text is in blue fonts. Please let me know when to upload our revised ms with all changes.

I hope that these changes in response to the reviewers are acceptable to you. We believe that with these changes, the re-revised version meets the requirements and you will kindly accord acceptance to our submission.

I look forward to your kind and early response.

With my Best Regards.

Yours sincerely,

N. Ramaiah Biological Oceanography Division CSIR-National Institute of Oceanography Dona-Paula, Goa, India

Responses to reviewer's comments The authors have used DNA sequence analysis of environmental 16S rRNA genes and phylogenetic reconstruction to investigate the community diversity of bacteria present at a time-series station in the Arabian Sea. Samples were obtained from the surface mixed layer and from within the oxygen minimum zone (OMZ) on three separate occasions over the course of several months and under contrasting hydrographic conditions. The authors conclude that while there was no distinct seasonal difference in community structure (lines 21-22), greater diversity and community richness were evident within the OMZ when compared to the surface and deep chlorophyll maximum. While the latter observation is of interest, it is based on a very modest number of sequences from each depth and time point (Table 5) and not that well supported by classical indices of diversity (Shannon and Simpson indexes). As the authors acknowledge at several points in the manuscript (e.g., Section 3.6, lines 334-335, lines 436-438, etc.) their findings are preliminary because due to the undersampling of the community they have not captured or analyzed in sufficient depth the far greater diversity evidently present at the station in order to draw robust conclusions.

We are grateful for these highlights you kindly offered on this submission. In view of your observations below, we have revised the ms to include the changes in accordance with your expert observations. Comment: Where they may be on firmer ground is in reporting that the composition of the surface community was distinct from that of the OMZ (Section 3.5). Much of this difference appears to be explained by the absence of cyanobacteria from the deeper samples (lines 195-198, lines 269-270), however, which is an expected result given their photoautotrophic nature. Indeed, if one ignores the contribution of cyanobacteria to the surface community, the relative percentage contributions of the dominant heterotrophs (alpha and gamma proteobacteria) at all depths would be much less distinct than that shown in Figure 2. Reanalyzing the data (minus cyanobacteria) might proof useful. Reply: We gratefully acknowledge this insightful observation on taking out the contribution of cyanobacteria from the surface community and looking at the contribution of Alphaproteobacteria and Gammaproteobacteria at all depths. Upon reanalyzing the data (minus cyanobacteria) the two following points were clear. 1- The percent contribution of Gammaproteobacteria increased by 9% in the surface and that Alphaproteobacteria by 4%. 2- Further the contribution of both Alphaproteobacteria and Gammaproteobacteria were in general lower in the surface than they were below DCM and in the core OMZ. Therefore, we may be permitted to retain our earlier statement as well as the figure 2 as they are.

The following are the replies for minor points. Comment: Line 13 What does 'Contributions" mean? Numbers, biomass, activity? Reply: This was meant as a general statement; in view of your advises, the sentence is suitably modified. Pl see lines 13-14 in the re-revised version. It reads: 'Role of microbial communities in terms of its biomass, number and activity in oxygen minimum oceanic zones are being realized through the applications of molecular techniques'

Comment: Line 84 What is 131 Tris? Reply: Sorry, It was just Tris and '131' is deleted on line 103 of the re-revised version of the manuscript.

Comment: Section 2.2 Much more detail of the PCR conditions is required. The reference to the manual of Sambrook (et al.) is insufficient - what were the temperatures, times, cycle numbers, etc.? Reply: Thank you so much. Complete details of PCR condition is now included in section 2.3, lines 118-120 in the re-revised manuscript. "DNA was extracted from Sterivex cartridges following the modified method of Ferrari and Hollibaugh (1999). The precipitated DNA was hydrated in 50 $\mu$l sterile deionised water, purified and quantified in a Nano-drop (Thermo Scientific, USA). All extracts were confirmed to be of PCR quality. Using the universal 16S rRNA primers, 27F and 1492R, the 16S rRNA gene was amplified as per the conditions given in Sambrook (1989). These conditions are: initial denaturation at 94°C for 4 min, 35 cycles consisting of denaturation at 94oC for 1 min, annealing at 55oC for 1 min and elongation at 72oC for 2 min and, final extension at 72oC for10 min. PCR amplification was performed in a final volume of 50 $\mu$l in a thermocycler (Applied Biosystems, USA) and correct amplification was ensured by checking for the amplicons electrophoretically."

Comment: Section 2.3 TA cloning is very efficient. What explains the poor numbers of transformants recovered in this study? Reply: We are unable to provide anymore explanation on the low transformation efficiency. The fact remains that we did multiple trails to achieve the clone numbers that we could garner.

Comment: Line 103 Manufacturers name and reaction conditions required Reply: As per the reviewer's suggestion, manufacturer's name and reaction condition is now stated in lines 133-135 in the re-revised version. "All positive clones/transformants from each sample were picked out, grown overnight at 37°C on LB plates and subjected to the colony-PCR with primers sets pucM13F/pucM13R using temperature conditions as per TOPO-TA cloning guide( Invitrogen): initial denaturation step of 10 min at 94°C, followed by 30 cycles of 94°C for 1 min, , annealing at 55°C for 1 min with elongation step at 72°C for 1 min and final extension at 72oC for10 mins."

Comment: Line 104-05 which primers and reaction conditions were used for sequencing? Reply: As per the suggestions, primers and reaction conditions used for sequencing is now mentioned in lines 138-140 in the re-revised version. "The PCR products

were purified with the Axyprep-96 PCR Clean up kit (Axygen, Biosciences) and then sequenced using 16S rRNA primers, 27F and 1492R in an ABI 3130XL genetic analyzer (Applied Biosystems, USA) with the temperature profile as follows: initial denaturation at 96°C for 1 minute, 30 cycles consisting of denaturation at 96oC for 10 sec, annealing at 55oC for 10 sec and elongation at 60oC for 4mins and final extension at 60oC for 1mins."